# Enteropathogenic *Providencia alcalifaciens*: A Subgroup of *P. alcalifaciens* That Causes Diarrhea

**DOI:** 10.3390/microorganisms12071479

**Published:** 2024-07-19

**Authors:** Dieter Bulach, Glen P. Carter, M. John Albert

**Affiliations:** 1Microbiological Diagnostic Unit Public Health Laboratory, Peter Doherty Institute for Infection and Immunity, The University of Melbourne, Melbourne, VIC 3000, Australia; dieter.bulach@unimelb.edu.au; 2Melbourne Bioinformatics, The University of Melbourne, Parkville, VIC 3010, Australia; 3Department of Microbiology and Immunology, Peter Doherty Institute for Infection and Immunity, The University of Melbourne, Melbourne, VIC 3000, Australia; glen.carter@unimelb.edu.au; 4Department of Microbiology, College of Medicine, Kuwait University, Jabriya P.O. Box 24923, Kuwait

**Keywords:** *P. alcalifaciens*, diarrhea, T3SS, enteropathogenicity, invasion, plasmid

## Abstract

Despite being considered a normal flora, *Providencia alcalifaciens* can cause diarrhea. In a previous study, strain 2939/90, obtained from a diarrheal patient, caused invasion and actin condensation in mammalian cells, and diarrhea in a rabbit model. Four Tn*phoA* mutants of 2939/90 produced negligible invasion and actin condensation in mammalian cells. Now, the parent strain and the mutants have been sequenced to locate Tn*phoA* insertion sites and determine the effect on virulence. A Tn*phoA* insertion was detected in the type three secretion system (T3SS) locus on a large plasmid and not in a T3SS locus on the chromosome. In 52 genomes of *P. alcalifaciens* surveyed, the chromosomal T3SS locus was present in all strains, including both *P. alcalifaciens* genomic clades, which we classified as group A and group B. Plasmid T3SS was present in 21 of 52 genomes, mostly in group A genomes, which included isolates from an outbreak of hemorrhagic diarrhea in dogs. The Tn*phoA* insertion only in the plasmid T3SS locus affected the invasion phenotype, suggested that this locus is critical for causation of diarrhea. We conclude that a subgroup of *P. alcalifaciens* that possesses this plasmid-mediated T3SS is an enteric pathogen that can cause diarrheal disease.

## 1. Introduction

*Providencia alcalifaciens* is a species in the *Providencia* genus of the family *Enterobacteriaceae* [1]. Since it is a lactose nonfermenting bacterium, it appears as pale colonies on enteric agars such as MacConkey agar, desoxycholate citrate agar, and *Salmonella-Shigella* agar, like other lactose nonfermenting bacteria such as *Salmonella*, *Shigella,* and *Proteus.* However, selective media have been developed for specific culturing of *P. alcalifaciens.* These include *P. alcalifaciens* medium (PAM) [2] and polymyxin-mannitol-xylitol medium for *Providencia* (PMXMP) [3]. By phenotypic tests, *P. alcalifaciens* can be differentiated from other species of *Providencia* [4]. *P. alcalifaciens* can be identified from its biochemical reactions using the commercial biochemical strip kit, API-20E (bioMerieux, Marcy-L’Etoile, France), and commercial automated systems such as Vitek-II (bioMerieux). Although *P. alcalifaciens* is considered a part of the normal flora of the feces, in many mammals, including humans, there is evidence to suggest that it can also cause diarrhea. It has been implicated in foodborne outbreaks of diarrhea [5,6] and travelers’ diarrhea [7]. In a case-control study of children’s diarrhea, the organism was isolated at a significantly higher rate from children with diarrhea than from matched control children [8]. It was reported to cause foodborne hemorrhagic diarrhea in dogs [9].

*P. alcalifaciens* strains are susceptible to thienamycin, ceftazidime, cefotaxime, ceftizoxime, and moxalactam. Other choices for antimicrobial therapy would include ceftriaxone, mezlocillin, imipenem, and trimethoprim-sulfamethoxazole [10]. They can be resistant to amoxicillin, ampicillin, erythromycin, tetracycline, and doxycycline [6]. *P. alcalifaciens* strains can produce inducible β-lactamases that will hydrolyze primary and extended-spectrum penicillins and cephalosporins [11]. For this reason, the susceptibility of *P. alcalifaciens* isolates needs to be tested.

Pathogenicity studies with diarrheal isolates suggested that *P. alcalifaciens* can invade cultured mammalian cells such as HEp-2 cells with actin condensation, and cause diarrhea in a reversible ileal tie adult rabbit diarrhea (RITARD) model with the invasion of the intestinal mucosa [12]. There were two modes of invasion of epithelial cells: one by endocytosis and the other through intercellular tight junction [13]. Tissue culture invasion was inhibited by an agent that prevented microfilament formation [14]. Further studies confirmed that many diarrheal isolates of *P. alcalifaciens* were invasive for mammalian cells, but some were not [15,16]. To define the basis of invasion, Tn*PhoA* mutagenesis of the diarrheal strain, *P. alcalifaciens* 2939/90, was used to demonstrate the effect on cell invasion and actin condensation. The Tn*PhoA* mutants exhibited negligible invasion and actin condensation in HEp-2 cell assays [17]. In the current study, the parent strain, 2939/90, and four Tn*PhoA* insertion mutants were sequenced to determine the insertion sites of Tn*PhoA* and elucidate the genetic basis of virulence, especially cell invasion. Evaluation of the distribution of genetic determinants contributing to cell invasion in strain 2939/90 and across the *P. alcalifaciens* species leads us to conclude that a specific lineage of *P. alcalifaciens* is diarrheagenic. We refer to this subgroup that causes diarrhea as enteropathogenic *P. alcalifaciens*.

## 2. Materials and Methods

### 2.1. Isolates

We studied the parent wildtype strain of *P. alcalifaciens* 2939/90, which was isolated from the rectal swab of a child with diarrhea who was dead on arrival at a hospital in Dhaka, Bangladesh. The strain grew as a pure culture on MacConkey agar, desoxycholate citrate agar, and *Salmonella-Shigella* agar. The identification was made using the biochemical strip API 20E (bioMerieux). This strain had an invasive phenotype for the intestine in an animal model of diarrhea and in an in vitro HEp-2 cell assay [8]. In addition, four Tn*PhoA* mutants of *P. alcalifaciens* 2939/90 (M-23, M-47, M-63, and M-78) that had negligible invasion in HEp-2 cells [9] were studied.

### 2.2. Genome Sequencing

A shotgun sequencing strategy was used for sequencing genomic DNA from the *P. alcalifaciens* 2939/90 and the four Tn*PhoA* mutants. Genomic DNA was extracted using the DNeasy blood and tissue extraction kit (Qiagen, Hilden, Germany). Sequencing libraries were prepared using the Nextera DNA sample preparation kit (Illumina, San Diego, CA, USA) and the sequence read data were produced on either the Illumina NextSeq (paired-end, 150 base reads) or MiSeq (paired-end, 300 base reads) instrument. Long-read shotgun sequence data for *P. alcalifaciens* 2939/90 genomic DNA were generated from a sequencing library prepared using the Rapid PCR Barcoding Kit and run on the Oxford Nanopore (ONT) MinION instrument (Didcot, UK). Sequence-read data for the mutants are available at the National Center for Biotechnology Information (NCBI) in Bioproject, PRJNA1073245, and the assembled closed genome sequence for *P. alcalifaciens* 2939/90 is available in NCBI Bioproject, PRJNA929094. The genome sequence was annotated at NCBI using Prokaryotic Genome Annotation Pipeline (PGAP) version 6.4.

### 2.3. Genome Assembly

A genome sequence was produced from the Illumina read data using Spades v3.9 [18] for each of the Tn*PhoA* mutants. A closed-genome sequence for *P. alcalifaciens* 2939/90 was assembled using dragonflye version 1.0.13 (https://github.com/rpetit3/dragonflye, accessed on 16 July 2024) on ONT long read data for the assembly; Illumina read data were used for correcting the ONT assembly using a read-mapping approach. In all cases, a preliminary purity check and confirmation of taxonomic classification was performed on read data sets using kraken2 (https://github.com/DerrickWood/kraken2, accessed on 16 July 2024; version 2.1.2) with the GTDB kraken2 database release 214 (generated by https://github.com/leylabmpi/Struo2 using https://gtdb.ecogenomic.org/, all accessed on 16 July 2024).

### 2.4. TnPhoA Insertion Site

The Tn*PhoA* insertion site(s) for each of the Tn*PhoA* mutants was determined using the first and the last 30 bases of the sequence of Tn*PhoA* (NCBI Accession, U25548.1) to screen for reads containing: CCGTTCAGGACGCTACTTGTGTATAAGAGTCAG (Bases 7701 to 7733, top strand of U25548) and TCCAGGACGCTACTTGTGTATAAGAGTCAG (Bases 1 to 30, reverse strand of U25548). The identified reads were aligned, and a consensus sequence was determined for each insertion site in each isolate.

### 2.5. Bioinformatics and Data Resources

Available *P. alcalifaciens* genome sequences from NCBI were obtained (downloaded on 1 February 2024). In total, 52 *P. alcalifaciens* genomes sequences were identified (including for strain 2939/90). The list comprises genomes identified as *P. alcalifaciens* on NCBI or GTDB (https://gtdb.ecogenomic.org/, accessed on 16 July 2024) and excluded metagenome assembled genomes and poor-quality genomes (i.e., from mixed isolates or incomplete genomes). Summary information about these genomes is presented in Appendix A. Type strain information was obtained from LPSN (https://lpsn.dsmz.de/, accessed on 16 July 2024) and a list of *P. alcalifaciens* genomes checked for genome completeness and taxonomic classification was obtained from GTDB. A comparison of genomic sequences using a k-mer based approach was conducted using Mashtree version 1.2.0 (https://github.com/lskatz/mashtree, accessed on 16 July 2024). Genome-wide average nucleotide identity (ANI) was determined using FastANI version 1.33 [19]. Phylogenetic trees were visualized using Figtree v1.4.4 (https://github.com/rambaut/figtree, accessed on 16 July 2024). Prediction of T3SS effector proteins was performed using EffectiveDB website accessed 13 October 2023 [20].

## 3. Results

### 3.1. Assembly and Annotation of P. alcalifaciens 2939/90

Assembly using dragonflye (v1.0.13) with Oxford Nanopore long- and Illumina short-read data produced a closed circular chromosome sequence (4,087,862 bp) and four circular plasmid sequences (p2939_90_1: 127,696; p2939_90_2: 84,555; p2939_90_3: 58,284; p2939_90_4: 7475). The genome sequence was annotated by NCBI (accession: GCF_029962585.1) using PGAP (version 6.6), which identified 4201 genes including 3970 protein coding genes and 121 pseudogenes located on the chromosome and plasmids.

### 3.2. Assembly and Characterization of the Genome Sequences of TnPhoA Mutants

Illumina read data were used to assemble genome sequences for each of the four Tn*PhoA* mutants of *P. alcalifaciens* strain 2939/90. Sequence and assembly information is presented in Appendix A. A survey of antimicrobial resistance genes showed the presence of additional resistance genes (*blaTEM-1*, *aph*(*6*)*-Ic*, *ble,* and *aph*(*3′*)*-IIa*) in each of the Tn*PhoA* mutants, all of which were not present in the parent strain 2939/90; this is consistent with the integration of Tn*PhoA* in the mutants. A core genome comparison of the parent strain and mutants showed the maximum distance between any isolate pair was two single nucleotide polymorphisms (SNPs), indicating a close genomic relationship between the mutants and *P. alcalifaciens* strain 2939/90 (Table 1).

### 3.3. TnPhoA Insertions

The location of Tn*PhoA* insertions in the genome of the Tn*PhoA* mutants was determined by searching for reads that contained the sequence (last 50 bases, see Section 2 for sequences) at either end of the Tn*PhoA* element. The locations are shown in Table 2 for each of the Tn*PhoA* mutants, with reference to the closed-genome sequence for *P. alcalifaciens* 2939/90 (Accession: GCF_029962585.1).

Each of the four Tn*PhoA* mutants carried two copies of Tn*PhoA*. Mutants M-23 and M-78 were isogenic. Each of the four Tn*PhoA* mutants had at least one Tn*PhoA* inserted in plasmid p2939_90_1, with Tn*PhoA* mutant M-63 having both Tn*PhoA* copies in p2939_90_1. The chromosomal Tn*PhoA* insertion in Tn*PhoA* mutant M-47 interrupts a gene related to fimbrial biosynthesis, while, for Tn*PhoA* mutants M-78 and M-23, the Tn*PhoA* insertion in plasmid p2939_90_4 interrupts a gene that may play a role in DNA conjugation. The insertion of Tn*phoA* in plasmid 1 of both M-78 and M-23 interrupts the gene-encoding pilotin (SctG).

### 3.4. TnPhoA Inserts in p2939_90_1 Have a Predicted Role in Type III Secretion

Plasmid p2939_90_1 is 127,696 bp and is predicted to encode 115 proteins. A locus extending from bp 119,200 to bp 127,696 and then from bp 1 through to bp 22,805, contains genes encoding proteins that are predicted to be part of a type III secretion apparatus. All Tn*PhoA* mutants contain at least one Tn*PhoA* insertion site in the region of p2939_90_1 (see Appendix A for predicted gene location and function along with the genes which were interrupted by Tn*PhoA* in specific mutants).

### 3.5. Two Type III Secretion Apparatus Loci in the P. alcalifaciens Strain 2939/90 Genome

Examination of the annotated chromosome of *P. alcalifaciens* strain 2939/90 identified another locus that is predicted to encode components of a type III secretion apparatus. The genes located in the respective type III secretion apparatus loci on the chromosome and on plasmid p2939_90_1 are shown in Table 3 (details of the location of the type III secretion apparatus genes on the chromosome are shown in Appendix A). 

### 3.6. Distribution of the Type III Secretion Apparatus Loci in P. alcalifaciens

Taking the 52 available *P. alcalifaciens* genome sequences, including the genome of strain 2939/90, we inferred genomic relationships among sequences using Mashtree (k-mer difference approach). A phylogenetic tree summarizing the inferred relationship is presented in Figure 1. We observed two major clades and identified these as Group A and Group B; strain 2939/90 is part of Group A. Using the region from 1,619,656 to 1,641,639 on the strain 2939/90 chromosome (type III secretion apparatus locus; see Appendix A) as query, we determined using Blast that this locus was present in each of Group A genome sequences. At a lower average nucleotide sequence identity (~85%), we detected a related locus in each of the Group B isolates; examination of the closed genome sequence for isolate 2019-04-29291-1-1 (Group B) showed a type III secretion apparatus locus with the same gene layout as for strain 2939/90 and with gene synteny in the regions flanking the locus between these genome sequences. The ANI between the Group A and Group B chromosomal genome sequence ranged between 88% and 89%; a similar rate of divergence between the Group A and Group B chromosomal type III secretion apparatus locus and the whole genome as well as genomic synteny suggested these chromosomally located type III secretion apparatus loci are orthologous and have been inherited vertically in *P. alcalifaciens*. 

Again, using Blast and this time using the type III secretion apparatus locus located on plasmid p2939_90_1 as query, we investigated the distribution of this locus among the 52 available *P. alcalifaciens* genome sequences. Nucleotide Blast showed that there was no significant nucleotide similarity between the chromosomal and plasmid-borne type III secretion apparatus locus; a broader search of the NCBI nr nucleotide database revealed that this locus is present in a sequence from *Providencia* and probably exclusively in *P. alcalifaciens*. Constraining the e-value to e-100, the chromosomal type III secretion apparatus locus was not detected and a near-identical locus was detected in a subset of the 52 genome sequences. Sequences containing the locus are shown in Figure 1. In total, 21 genome sequences contained the locus, with most being Group A genome sequences (17/21).

### 3.7. Type III Secreted Effector Protein Prediction

A survey of type III secreted effector proteins encoded on the *P. alcalifaciens* strain 2939/90 was performed using EffectiveDB; a summary of the number of effector proteins predicted to be encoded on each replicon is shown in Table 4. A total of 26 predicted effectors were encoded on p2939_90_1. Among the effector proteins on this plasmid predicted using EffectiveDB (see Appendix A), there were three effectors that would be predicted by protein similarity to a characterized effector protein (PO864_RS19515, related to the IpaC/SipC family of effector proteins; PO864_RS20070, related to IacP family of effector proteins; and PO864_RS19620, related to BopA family of effector proteins); however, most predicted effector proteins were classified as hypothetical by protein sequence similarity. The distribution of these predicted effector genes among the 21 genomes carrying the p2939_90_1 T3SS locus is summarized in Appendix A and shown in detail in Appendix A.

## 4. Discussion

Four Tn*PhoA* mutants of *P. alcalifaciens* strain 2939/90 were previously characterized and shown to have negligible invasion and actin condensation in HEp-2 cells [17]. Genomic characterization of these mutants has shown that these mutants have genomes that are highly related to the parental *P. alcalifaciens* 2939/90 strain (two or fewer core genome SNPs) and that the change in antimicrobial resistance genotype in the mutants is consistent with the insertion of Tn*PhoA* element into a replicon in each of the mutants. Interestingly, we determined that there were two Tn*PhoA* elements inserted in each of the mutants. While this is at odds with the observations made by [17], where mutants were reported to contain a single insertion, additional insertion would have occurred during the subsequent propagation of the mutants.

Determination of the insertion sites showed that mutants M-78 and M-23 had identical Tn*PhoA* insertions (Table 2) and are therefore isogenic mutants. Each of the mutants had a Tn*PhoA* on p2939_90_1 except M-63, which had two Tn*PhoA* insertions on p2939_90_1. The remaining insertion sites were located on the chromosome or p2939_90_4. Based on the similarity of the cell culture phenotype seen in all four mutants, it could be assumed that the same function was being impacted by the Tn*PhoA* insertions in each of the mutants. Examination of the insertion points on p2939_90_1 and the predicted genes encoded at the points of insertion showed that all insertions on p2939_90_1 interrupted the genes involved in the type III secretion and all these genes were part of a type III secretion apparatus locus (Table 2).

A type III secretion apparatus locus was also found on the chromosome of strain 2939/90; an orthologous locus was found on all 52 sequenced *P. alcalifaciens* genomes. This locus is likely to produce a type III secretion apparatus that is independent of the type III secretion apparatus encoded by the locus on p2939_90_1. This is supported by the observation that the p2939_90_1 locus was not present in all sequenced *P. alcalifaciens* genomes, and its distribution is not monophyletic (see Figure 1), potentially indicating horizontal movement of this locus. In this plasmid carrying the T3SS locus, we found the presence of three insertion sequences (ISs)—IS*3*, IS*200*/IS*605*, and IS*481*—any of which could be involved in the horizontal transfer of this locus. Both the chromosomal and plasmid p2939_90_1 loci contain all the required components to form a type III secretion apparatus [21], with the p2939_90_1 locus containing some additional, non-essential genes for the formation of a type III secretion apparatus. It appears that the T3SS locus on chromosome does not contribute to virulence in humans. Moreover, we predict that as many as 26 effector proteins are encoded by p2939_90_1, which are likely to require the apparatus encoded on the p2939_90_1 for secretion. By protein similarity, several proteins that are effectors in other pathogenic bacteria were detected in *P. alcalifaciens.* These included IagB, IacP/SinF, IpaC/SipC, BopA, EspG domain-containing protein, and TcdA/TcdB catalytic glycosyltransferase domain-containing protein. IagB and IacP/SinF are invasion proteins in salmonella [22,23], and IpaC/SipC is involved in invasion of epithelial cells by shigella/salmonella [24]. The presence of these three invasive proteins in *P. alcalifaciens* strain 2939/90 is enough proof of their roles in invasive diarrhea. BopA is an effector protein secreted by *Burkholderia pseudomallei* via the type III secretion system, and it has been shown to play a crucial role in the escape of the bacterium from autophagy [25]. EspG is an effector protein shared by enteropathogenic *Escherichia coli*, enterohaemorrhagic *E. coli,* and shigella. It causes microtubule destabilization and cell detachment [26]. TcdA and TcdB are primary virulence factors of *Clostridium difficile.* They enter and disrupt host cell function by glucosylating and thereby inactivating key signaling molecules within the host [27]. Characterization of the function of the predicted effector proteins on p2939_90_1 seems to be a logical next step to investigate the complete diarrheagenic properties of *P. alcalifaciens.*

Thus, *P. alcalifaciens* 2939/90 carried two type III secretion systems (T3SSs). There are other pathogenic bacteria that are known to harbor more than one T3SS. These include *Salmonella enterica* [28], *Yersinia enterocolitica* [29], enterohaemorrhagic *Escherichia coli* O157:H7 [30], *Vibrio parahaemolyticus* [31], and *Burkholderia pseudomallei* [32,33]. Both T3SSs were reported to be functional in *S. enterica* [29] and *V. parahaemolyticus* [31]. The invasive phenotype associated with strain 2939/90 in cell culture and the circumstances of the isolation of this strain from a fatal human case of diarrhea strongly suggest that *P. alcalifaciens* lineages carrying the p2939_90_1 type III secretion apparatus locus are likely to be able to cause severe diarrheal disease. While causation is difficult to demonstrate in humans, *Canis lupus familiaris* (dog) is a host that may be useful for the demonstration of causation. We note that the *P. alcalifaciens* isolates characterized in an outbreak and associated with acute hemorrhagic diarrhea in dogs [9] carried the p2939_90_1 type III secretion apparatus locus (shown in Figure 1).

While *P. alcalifaciens* is a well-recognized part of the normal flora of many animals, including humans, it is not unprecedented for certain lineages within a commensal species to cause severe diarrheal disease; a case in point is diarrheagenic *E. coli.* Even though *E. coli* is a commensal flora, at least five subgroups are recognized as primary diarrheal pathogens [34]. Similarly, *P. alcalifaciens* strains that possess a p2939_90_1 type plasmid that carries a type III secretion apparatus locus are likely to be diarrheagenic. Such strains can be considered enteropathogenic as opposed to non-pathogenic normal flora strains.

## 5. Conclusions

This work identified a T3SS encoded on p2939_90_1 (encoding both secretion apparatus and effector proteins) that may act independently of chromosomally encoded T3SS. The p2939_90_1 encoded T3SS contributes to the invasion phenotype observed in *P. alcalifaciens* strain 2939/90. The characterization of *P. alcalifaciens* strain 2939/90 and the observation that the presence of the p2939_90_1 T3SS in other isolates is associated with diarrheal disease is a significant step towards recognizing that a lineage or subgroup of *P. alcalifaciens* is a causative agent of diarrheal disease. This enteropathogenic lineage should be included in diarrheal disease investigations. The identification of genetic determinants that play a central role in disease causation in *P. alcalifaciens* makes it feasible to differentiate diarrhea-causing lineage from normal flora lineages. A unique sequence that may be present in the pathogenic locus (p2939_90_1 T3SS locus) may be useful in a PCR assay to detect enteropathogenic strains of *P. alcalifaciens*.

Our journey of the discovery of a subgroup of *P. alcalifaciens* as a causative agent of diarrhea has been an interesting one, as outlined in the Introduction. We first had the clinical observation of a child with severe diarrhea who died and from whom a pure culture of the bacterium was grown; we reproduced diarrhea in a rabbit model, demonstrated an invasive mechanism of diarrhea by examining the intestine of the infected animal model and in an in vitro cell culture model, and abrogated the cellular invasion by Tn*phoA* mutagenesis. Through the current genomic sequencing study of the parent strain and its Tn*phoA* mutants, we found evidence that a plasmid-borne T3SS is the basis of the pathogenicity of diarrheagenic *P. alcalifaciens.*

## Figures and Tables

**Figure 1 microorganisms-12-01479-f001:**
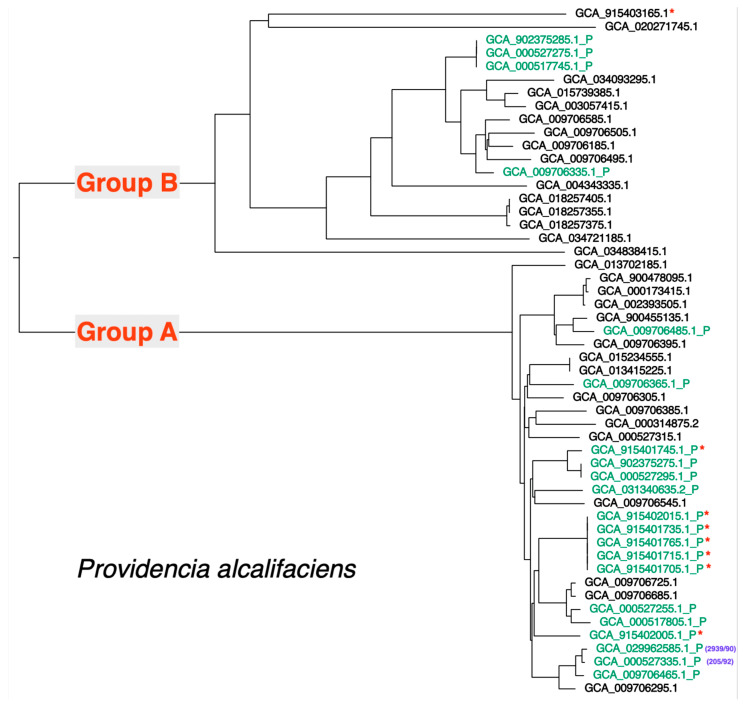
A phylogenetic tree showing the inferred relationship among the 52 available *P. alcalifaciens* genome sequences. The relationship was inferred using Mashtree. The tree shows two main clades of isolates (labelled Group A and Group B). Genome sequences were identified by the GenBank assembly accession numbers. Genome sequences containing the p2939_90_1 type three secretion system have a “_P” suffix and the taxon label is colored green. Sequences that were part of the Norwegian *P. alcalifaciens* outbreak in dogs are identified with a red asterisk [9]. Strains 2939/90 and the related strain 205/92 [14] are identified with a purple text.

**Table 1 microorganisms-12-01479-t001:** Core genome, pairwise SNP distance between parental strain (2939/90) and Tn*PhoA* mutants.

Strain/Mutant	2939/90	M-23 *	M-47 *	M-63 *	M-78 *
2939/90	0	1	2	1	1
M-23	1	0	1	0	0
M-47	2	1	0	1	1
M-63	1	0	1	0	0
M-78	1	0	1	0	0

* Tn*PhoA* mutants [17].

**Table 2 microorganisms-12-01479-t002:** Tn*PhoA* insertion sites in mutant isolates of *P. alcalifaciens* 2939/90.

Site	Left	Right	Replicon	Gene(s)	Tn*phoA* Mutant Isolate
Site 1	>site1 GCTTTGTTAGCACTAGCCAAAAAAC ATGGTTGGTCATTATCGAGAGAAAT	>site1 TAACAAAGCATCTTTCTGTTCTTTT GTTAAATAAATAAAAGCTCTTTCGT	p2939_90_4	PO864_20965 to PO864_20995 (hypothetical protein)	M-78, M-23
Site 2	>site2 GCATAGATGAATGCTTGAATTATTT AGACATGAGTACATTAGGCAAACAA	>site2 CATCTATGCTTGTTGTTCTATCCTC AGAGCTAATAGATTTGTTGAGACTC	p2939_90_1	PO864_RS20065 (SctG)	M-78, M-23
Site 3	>site3 CATTAAAGGTTACGAAAGAAGGCGG AGCTGATATTACCGATGACCCGTTA	>site3 CCTTTAATGCAACATCATTCGCATC AAACATTAACGGCTCAACAATCTCT	p2939_90_1	PO864_19795 (SctC)	M-63
Site 4	>site4 ACGCAACACCAACCGCGGCCAATGC TAAACTCGCACCACCAGAGAATACG		p2939_90_1	PO864_19725 (SctE)	M-63
Site 5	>Site5 GTGTAATGGTGAACAATACGGGTGT TGATTTACTACCCACATTGGGTAAA		2939/90 chromosome	PO864_08195 (fimbrial protein)	M-47
Site 6	>site6 ACTGAGCCTCGCTTCGTTCAGATAA AACAACAGAAATATTATCATATTCA		p2939_90_1	PO864_20250 (SctJ)	M-47

**Table 3 microorganisms-12-01479-t003:** Gene-encoding type III secretion apparatus proteins and some effectors in strain 2939/90.

Chromosome	Unified Nomenclature ^	Plasmid p2939_90_1
PO864_RS07710	SctA	PO864_RS19510
PO864_RS07705	SctB	PO864_RS19515
PO864_RS07700	SctE	PO864_RS19520
PO864_RS07695	chaperone	PO864_RS19525
PO864_RS07690	SctU	PO864_RS19530
PO864_RS07685	SctT	PO864_RS19535
PO864_RS07680	SctS	PO864_RS19540
PO864_RS07675	SctR	PO864_RS19545
PO864_RS07670	SctQ	PO864_RS19550
PO864_RS07665	SctP	PO864_RS19555
PO864_RS07660	SctO	PO864_RS19560
PO864_RS07655	SctN	PO864_RS19565
PO864_RS07650	chaperone	PO864_RS19570
PO864_RS07645	SctV	PO864_RS19575
PO864_RS07640	SctW	PO864_RS19580
PO864_RS07635	SctC	PO864_RS19585
PO864_RS07630	regulator	PO864_RS19590
Absent		PO864_RS20070 ^a^
Absent	SctG	PO864_RS20065
Absent		PO864_RS20060 ^b^
Absent		PO864_RS20055 ^c^
Absent		PO864_RS20050 ^d^
PO864_RS07625	SctD	PO864_RS20045
PO864_RS07620	SctF	PO864_RS20040
PO864_RS07615	SctI	PO864_RS20035
PO864_RS07610	SctJ	PO864_RS20030
PO864_RS07605	SctK	PO864_RS20025
PO864_RS07600	SctL	PO864_RS20020
PO864_RS07595	Hypothetical protein	Absent
Absent	Effector (IpaC/SipC)	PO864_RS20015
Absent	Effector (BopA)	PO864_RS19620

^ Using the nomenclature proposed by [21]; ^a^ iacP/sipF: helps with invasion in salmonella; ^b^ transcriptional regulator of virulence genes in salmonella (HilA/EilA); ^c^ type III secretion system invasion protein, IagB in salmonella; ^d^ hypothetical protein. SctG is pilotin that stabilizes export apparatus. It is a lipoprotein that assists the formation of secretin ring in the outer membrane of type three secretion system.

**Table 4 microorganisms-12-01479-t004:** Density of predicted T3SS effector CDS on each replicon of *P. alcalifaciens* strain 2939/90.

Replicon	T3SS Effector (High)	T3SS Effector (Low)	Total CDS	Percent of CDS Predicted to Containt3ss Effectors
Chromosome	427	209	3580 ^	17.8%
p2939_90_1	20	6	89 ^	29.2%
p2939_90_2	5	3	73	11.0%
p2939_90_3	3	4	47	14.9%
p2939_90_4	0	0	8	0.0%

^ Genes encoding type III secretion apparatus not included in the count.

## Data Availability

The original contributions presented in the study are included in the article/Appendix A, further inquiries can be directed to the corresponding author.

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
