# Peer review of "Enteropathogenic Providencia alcalifaciens: A Subgroup of P. alcalifaciens That Causes Diarrhea"

_microorganisms, 2024, doi:10.3390/microorganisms12071479_

Round 1

Reviewer 1 Report

Comments and Suggestions for Authors

This study is potentially interesting and contains important findings. However, there’s still a few significant drawbacks, which in detail is as below. Thus, my suggestion is substantial revision.

Firstly and most importantly, this study has based on only genome analysis. The authors should consider using other techniques, such as RNAseq or gene expression, or conjugation/transformation (to analyze the efficiency), etc., to further elucidate and validate their observation and findings. Since the manuscript touches upon diarrhea, then expression of virulence should be important and play a key role here. I reckon the authors need to show more convincing results, especially on the gene expression level, protein level, etc. Without this, the conclusion looks less convincing.

Secondly, the authors should clarify on what criteria or base they stated the importance of locus. For example, suppose the gene is located on the plasmid, however, what backbone of such plasmid is. Cause, the context upstream or downstream of the plasmid is essential, as this leads to the likeliness of the spread of genes. For another example, IS26 is an important gene element that plays a key role in horizontal transfer.

Thirdly, for the genome sequencing, please make the description more clear, as it’s a little bit confusing in a few questions as:  Did the authors perform both NanoPore and Illumina? If so, should base on the long read from NanoPore, then correct using Illumina data? Or use the NanoPore long read to assemble the contigs from Illumina? This could cause bias or missing of repeats.

The English writing needs to be improved.

Comments on the Quality of English Language

The English needs to be improved.

Author Response

Referee 1

This study is potentially interesting and contains important findings. However, there’s still a few significant drawbacks, which in detail is as below. Thus, my suggestion is substantial revision.

Firstly and most importantly, this study has based on only genome analysis. The authors should consider using other techniques, such as RNAseq or gene expression, or conjugation/transformation (to analyze the efficiency), etc., to further elucidate and validate their observation and findings. Since the manuscript touches upon diarrhea, then expression of virulence should be important and play a key role here. I reckon the authors need to show more convincing results, especially on the gene expression level, protein level, etc. Without this, the conclusion looks less convincing.

Response. Thank you for the comments. We first had the biological observations in a series of publications described in the Introduction. In the current genomic study, we sought the genetic basis of our biological observations. We have now summarized this in Conclusions as “ Our journey of the discovery of a subgroup of P. alcalifaciens as a causative agent of diarrhea has been an interesting one as outlined in the Introduction. We first had/established the biological observation/experiments – isolation of the bacterium as a pure culture from the rectal swab of a dead child with diarrhea, reproduction of diarrhea in a rabbit model of diarrhea, establishment of an invasive mechanism by studying the intestine of infected animals by light and electron microscopy and by studying the cell culture model of infection, and abrogation of invasion of the bacterium by TnphoA mutagenesis. Through the current genomic sequencing study of the parent strain and its TnphoA mutants, we found evidence that a plasmid-borne T3SS is the basis of the pathogenicity of diarrheagenic P. alcalifaciens.”  (L164-173).

Secondly, the authors should clarify on what criteria or base they stated the importance of locus. For example, suppose the gene is located on the plasmid, however, what backbone of such plasmid is. Cause, the context upstream or downstream of the plasmid is essential, as this leads to the likeliness of the spread of genes. For another example, IS26 is an important gene element that plays a key role in horizontal transfer.

Response. Thank you for the comment. We have now added the following. “In this plasmid carrying the T3SS locus, we found the presence of three insertion sequences (ISs) – IS3, IS200/IS605, and IS481 – any of which could be involved in the horizontal transfer of this locus.” (L105-107).

Thirdly, for the genome sequencing, please make the description more clear, as it’s a little bit confusing in a few questions as:  Did the authors perform both NanoPore and Illumina? If so, should base on the long read from NanoPore, then correct using Illumina data? Or use the NanoPore long read to assemble the contigs from Illumina? This could cause bias or missing of repeats.

Response. For the assembly of the closed genome sequence for strain 2939/90 we have used both Illumina and Nanopore read sets. It is clearly stated in the Materials and Methods in Section 2.3 that: “P. alcalifaciens 2939/90 was assembled using dragonflye on ONT long read data for the assembly; Illumina read data were used for correcting the ONT assembly using a read mapping approach.” We believe no further clarification is necessary.

The English writing needs to be improved.

Response. There were a few minor mistakes. These have been corrected.

Reviewer 2 Report

Comments and Suggestions for Authors

This is an interesting study pointing out a novel pathogen causing diarrhea, Providencia alcalifaciens. The paper is well written and the results are well founded. A few remarks to be addressed by the authors:

(a) The authors could mention in their introduction some info about the microbiological, biochemical properties of Providencia alcalifaciens, since it is an emerging pathogen and an average reader especially clinical microbiologists are not familiar with this bacterial species. Also, they could mention available treatment options if needed as well as data regarding antimicrobial resistance. 

(b) Authors should also mention in their material and methods how this bacterial isolated (was it from stool sample or blood culture), what type of culture media were used, how this bacterial isolate was identified.

(c) Regarding the phylogenetic analysis, the authors should mention which method was employed in the calculation of genetic distances and whether they verified the phylogenetic trees statistically (i.e. bootstrapping). 

(d) Language needs minor editing since there are some grammatical errors and typos. 

Comments on the Quality of English Language

The paper is well written. Some minor comments need to be addressed. 

Author Response

Referee 2

This is an interesting study pointing out a novel pathogen causing diarrhea, Providencia alcalifaciens. The paper is well written and the results are well founded. A few remarks to be addressed by the authors:

(a) The authors could mention in their introduction some info about the microbiological, biochemical properties of Providencia alcalifaciens, since it is an emerging pathogen and an average reader especially clinical microbiologists are not familiar with this bacterial species. Also, they could mention available treatment options if needed as well as data regarding antimicrobial resistance.

Response. Since it is a lactose nonfermenting bacterium, it appears as pale colonies on enteric agars such as MacConkey agar, desoxycholate citrate agar, and Salmonella-Shigella agar like other lactose nonfermenting bacteria such as Salmonella, Shigella and Proteus. However, selective media have been developed for easy differentiation of P. alcalifaciens. These include P. alcalifaciens medium (PAM) (Senior, 1997) and polymyxin-mannitol-xylitol medium for Providencia (PMXMP) (Yoh et al, 2005). By phenotypic tests, P. alcalifaciens can be differentiated from other species of Providencia (O’Hara et al, 2000). P. alcalifaciens can be identified from its biochemical reactions using the commercial biochemical strip kit, API-20E (bioMerieux) and the commercial automated systems such as Vitek-II (bioMerieux). (L29-38).

  1. alcalifaciens strains are susceptible to thienamycin, ceftazidime, cefotaxime, ceftizoxime, and moxalactam. Other choices for antimicrobial therapy would include ceftriaxone, mezlocillin, imipenem, and trimethoprim-sulfamethoxazole (Spach and Liles, 1999). They can be resistant to amoxicillin, ampicillin, erythromycin, tetracycline and doxycycline (Shah etal, 2015). P. alcalifaciens strains can produce inducible β-lactamases that will hydrolyze primary and extended-spectrum penicillins and cephalosporins (Swenson et al, 1999). For this reason, the susceptibility of P. alcalifaciens isolates needs to be monitored. (L45-51).

(b) Authors should also mention in their material and methods how this bacterial isolated (was it from stool sample or blood culture), what type of culture media were used, how this bacterial isolate was identified.

Response. We studied the parent wildtype strain of P. alcalifaciens 2939/90 that was isolated from the rectal swab of a child with diarrhea who was dead on arrival at a hospital in Dhaka, Bangladesh. The strain grew as a pure culture on MacConkey agar, sodium desoxycholate agar, and Salmonella-Shigella agar. The identification was made using the biochemical strip API 20E (bioMerieux). (L72-76).

(c) Regarding the phylogenetic analysis, the authors should mention which method was employed in the calculation of genetic distances and whether they verified the phylogenetic trees statistically (i.e. bootstrapping). 

Response. We have provided details of the method used to infer to tree in section 2.5 of the Materials and Methods. Mashtree (a k-mer difference method for inferring relationships between genome sequences) was used in to enable incorporation of all available validated P. alcalifaciens genome sequences (by GTDB) in our analysis. The key relationship we rely on in our analysis (i.e. Group A and Group B) has been confirmed by evaluation of the average nucleotide analysis and comparison of the chromosomal T3SS locus.

(d) Language needs minor editing since there are some grammatical errors and typos. 

Response. Language has been edited.

Round 2

Reviewer 1 Report

Comments and Suggestions for Authors

The authors have made substantial revision on this manuscript, and have addressed my comment accordingly.

One minor comment is, the info in Table 1 makes very few sense. I suggest the authors included such info (actually from 2 previous references) into the text, and then remove Table 1.

Comments on the Quality of English Language

The English writing meets the standard.

Author Response

Thank you for the comment regarding Table 1. This Table has been deleted now, and the other Tables have been renumbered accordingly. Kind regards.